# Maternal Obesity and Gut Microbiota Are Associated with Fetal Brain Development

**DOI:** 10.3390/nu14214515

**Published:** 2022-10-27

**Authors:** Sanjay Basak, Ranjit K. Das, Antara Banerjee, Sujay Paul, Surajit Pathak, Asim K. Duttaroy

**Affiliations:** 1Molecular Biology Division, ICMR-National Institute of Nutrition, Indian Council of Medical Research, Hyderabad 500007, India; 2Department of Health and Biomedical Sciences, University of Texas Rio Grande Valley, Brownsville, TX 78520, USA; 3Chettinad Academy of Research and Education (CARE), Chettinad Hospital and Research Institute (CHRI), Department of Medical Biotechnology, Faculty of Allied Health Sciences, Chennai 603103, India; 4Tecnologico de Monterrey, School of Engineering and Sciences, Campus Queretaro, Av. Epigmenio Gonzalez, No. 500 Fracc. San Pablo, Queretaro 76130, Mexico; 5Department of Nutrition, Institute of Basic Medical Sciences, Faculty of Medicine, University of Oslo, 0316 Oslo, Norway

**Keywords:** obesity, pregnancy, microbiota, placenta, brain development, fetal development, maternal obesity

## Abstract

Obesity in pregnancy induces metabolic syndrome, low-grade inflammation, altered endocrine factors, placental function, and the maternal gut microbiome. All these factors impact fetal growth and development, including brain development. The lipid metabolic transporters of the maternal-fetal-placental unit are dysregulated in obesity. Consequently, the transport of essential long-chain PUFAs for fetal brain development is disturbed. The mother’s gut microbiota is vital in maintaining postnatal energy homeostasis and maternal-fetal immune competence. Obesity during pregnancy changes the gut microbiota, affecting fetal brain development. Obesity in pregnancy can induce placental and intrauterine inflammation and thus influence the neurodevelopmental outcomes of the offspring. Several epidemiological studies observed an association between maternal obesity and adverse neurodevelopment. This review discusses the effects of maternal obesity and gut microbiota on fetal neurodevelopment outcomes. In addition, the possible mechanisms of the impacts of obesity and gut microbiota on fetal brain development are discussed.

## 1. Introduction

Obesity during pregnancy is a rising public health concern rapidly increasing worldwide [1,2,3]. Excessive maternal weight gain during pregnancy is consistently associated with many adverse impacts, including neurocognitive outcomes in the offspring [1,4]. Recently, the effect of obesity in pregnancy on maternal and fetal health has been reviewed [5,6]. The adverse effect of maternal obesity on the fetal programming of adult diseases extends beyond non-communicable diseases to brain diseases [7,8]. Obesity and a high-fat diet predispose offspring to adverse cardiometabolic and neurodevelopmental outcomes [9]. In addition, maternal high-fat intake during pregnancy was associated with an elevated risk of neuropsychiatric disorders such as hyperactivity/attention-deficit disorder/anxiety and depressive-like behaviours later in life in offspring [10,11,12,13,14,15,16,17,18]. Both obesity and a high-fat diet can impact maternal lipid metabolic state and gut microbial composition and affect offspring’s metabolic health and brain development [19,20]. Available data from extensive epidemiologic studies indicate an association between maternal obesity and adverse neurodevelopmental outcomes in human offspring. Maternal diet, adiposity, peripheral inflammation, and gut microbiota are the potential mechanisms for underlying changes in offspring brains and behaviour [20,21,22,23].

The diversity of intestinal microbiota during pregnancy has emerged as an essential factor as their metabolites can affect the host’s health in multiple ways, from lipid metabolism to brain development and function. The microbiota-gut axis can regulate maternal obesity, diet, lifestyle and physical activity. Gut microbiota produces neurotransmitters and neuromodulators (e.g., serotonin, GABA, SCFAs and their metabolites) and their derivatives in the circulation. These bioactive factors are transported to the brain via blood vessels after crossing the BBB and modulate the neonate’s cognitive development and brain-mediated performance activities. A steady-state microbiota of the mother is determined by several extrinsic (environment, geography, lifestyle) and intrinsic (genetics, mode of delivery, eating habits, age, infection, stress, medication) factors. While one-third of the gut microbial composition is common in most individuals, the remaining two-thirds are specific to the individual [24]. Despite these, not much data is available on maternal obesity during pregnancy and the modulation of gut microbial composition and diversity on the brain development of the offspring.

In this review, we discuss the impacts of maternal obesity and gut microbiota on offspring brain functionality and development. We also highlight underlying mechanisms such as maternal metabolic state, maternal and fetal inflammation, placental lipid transport, maternal gut microbiome, epigenetic modifications of neurotrophic genes, and impaired serotonergic and dopaminergic signalling. 

## 2. Impacts of Obesity on Maternal Endocrine Factors and Fetal Brain Development

In pregnancy, the maternal metabolism changes dramatically to support fetal growth and meet maternal additional energy requirements. However, maternal obesity in pregnancy increases maternal complications and enhances the risk of developing obesity, cardiovascular disease, diabetes, and cognitive dysfunction in offspring in adult life [25,26]. Obesity-induced metaflammation results in aberrant changes at the cellular and humoral levels that contribute to pregnancy complications. However, the physiological inflammatory process favours implantation, placental development, and parturition [27]. Complex interactions mediate adverse effects of obesity between metabolic, inflammatory, and oxidative stress homeostasis. Obesity in pregnancy is associated with elevated levels of pro-inflammatory cytokines such as interleukin 6 (IL-6), IL-1β, IL-8, and monocyte chemotactic protein-1 (MCP-1) in both the placenta and in maternal plasma [28,29]. The maternal obesity-induced swing toward a pro-inflammatory state can affect the neurometabolic state of the fetus.

Obesity-induced changes in maternal hormone levels can affect gene expression involved in fetal brain growth and development [30,31]. Obese pregnant women possess an increased risk of developing thyroid dysfunction during gestation. Maternal hormones such as thyroid and glucocorticoids affect fetal brain development. The hormones’ effects on brain development are time- and concentration-dependent [32]. Since obesity is associated with hypothyroidism and consequently affects brain development [33]. Maternal thyroid metabolism is also disturbed by maternal iodine deficiency, environmental endocrine modifiers, and other intrinsic factors associated with thyroid diseases [34]. 

In early pregnancy, thyroid hormones are needed for brain developmental processes such as neuronal migration, differentiation of neurons, glial cells, neurogenesis, and synaptogenesis [35,36]. Thyroid hormones regulate gene expression in the cell cycle and intracellular signalling, cytoskeleton organization, extracellular matrix proteins and several cellular adhesion factors involved in neuronal migration and neurogenesis [35,37]. In addition, thyroid hormones are also involved in neuronal migration and neuron development by regulating reelin and neurogenin 2 genes [37]. Therefore, optimum maternal thyroid function is critically required for fetal neurodevelopment. 

The altered thyroid hormone levels result in severe neurological deficiency and mental disability [38]. In addition, thyroid deficiency during pregnancy may drive offspring to the later onset of neurodevelopmental disorders [34,39,40]. Several neurodevelopmental deficits are observed in offspring born from mothers with thyroid dysfunction during the first half of gestation [41,42]. Even babies of the mother with asymptomatic thyroid dysfunction are at increased risk of impaired brain development [43]. The adverse subclinical hypothyroidism on fetal neurocognitive development is less specific. Maternal hypothyroxinemia occurs in mild iodine deficiency and may result in neurodevelopmental problems. 

Corticosteroids in late gestation are indispensable for fetal brain maturation [44]. The developing brain is susceptible to corticosteroid-induced stress [45]. A study showed that prenatal maternal stress increased glucocorticoids, and a high-fat diet, increased the risk of developing obesity in offspring in later life [46]. Chronic activation of glucocorticoid receptors alters the levels of glucocorticoid in hippocampal and hypothalamic regions, thus modifying feedback regulation of the hypothalamic-pituitary-adrenal (HPA) axis [47]. Corticosteroid effects are also mediated via epigenetic changes in genes associated with synaptic plasticity [48]. 

Stress hormones such as catecholamines, vasopressin, and oxytocin affect fetal brain development and functionalities [49]. Chronically increased cortisol levels in maternal stress result in lower levels of maternal T4 available for the fetal brain, thus affecting fetal brain development [50]. 

In addition to thyroid and glucocorticoid, several other hormones play roles in fetal neurodevelopment [51]. Maternal obesity and abnormal hormone levels affect fetal programming beyond the endocrine and cardiovascular systems to the brain. Several studies showed an association between high maternal BMI and adverse neurodevelopmental functions in their progenies (Table 1). 

## 3. The Placenta of an Obese Mother and Its Impact on Fetal Brain Development 

An increase in total lipid content characterizes the placentas of obese women at term, infiltrated neutrophils, foam-loaded macrophages and increased levels of pro-inflammatory mediators [68,69]. The maternal obesity induced-metabolic changes affect early placental growth, gene expression, and subsequent placental structure and function, which becomes clinically manifest in late pregnancy [70]. Placental dysfunctions may adversely affect fetal development [71]. In early pregnancy, the human placenta also responds to elevated maternal insulin in obese women. Obesity in pregnancy affects human placental structure and function in many ways. The cellular signalling system may mediate these effects by modulating inflammation, metabolism, and oxidative stress pathways. These placental alterations affect pregnancy outcomes independently and synergistically with other risk factors [6,72]. The placenta is enriched with a complex vascularization for fetal blood supply that requires extensive angiogenesis. Sub-optimal angiogenesis leads to abnormal placental size and vasculature. Dysregulated angiogenesis in the placenta may directly or indirectly involve pregnancies, including pre-eclampsia, pre-term birth [73], GDM [74], and IUGR [75]. The n-3 fatty acids deficiency reduced the placental transfer of fatty acids in pre-eclampsia and GDM -associated fetuses [76]. Abnormal placental vasculature is present in several pathological conditions, such as pre-term birth, intrauterine growth restriction (IUGR), and pre-eclampsia. Suboptimal placental angiogenesis is contributed by genetics, dietary and lifestyle factors [77]. Optimal placentation is facilitated by several angiogenic factors such as vascular endothelial growth factor A (VEGFA), angiopoietin-like 4 (ANGPTL4), fibroblast growth factor (FGF), and placental growth factor (PlGF), as well as docosahexaenoic acid, 22:6 n-3 (DHA) [78,79]. 

High-fat diets and maternal obesity alter the metabolome and early changes in the placental transcriptome and decrease placenta vascularity [80]. During pregnancy, a maternal high-fat diet promotes ectopic lipid deposition, leading to lipotoxicity and chronic inflammation in the placenta [81]. Further, the high-fat diet enforces the placenta to adapt its metabolic response and structural change (thickness) by altering angiogenesis. Animal studies showed reduced placental labyrinth depth and elevated expression of insulin-like growth factor 2 (IGF2) and its receptor genes in the fetuses of high-fat diet dams [82]. The n-3 polyunsaturated fatty acids (n-3 PUFA) deficiency resembles high-fat diet-induced impaired placental phenotypes. The maternal n-3 PUFA deficiency affected the vascular development of decidua; the feto-placental unit suggests impacts of maternal fatty acid status on placental vascularity [83]. 

The lipid accumulation in the obese placenta results from altered activities of fatty acid transporter expression, lipoprotein lipase, and alterations to mitochondrial oxidative metabolism [84,85]. A recent analysis of the genome-wide transcriptome, epigenetics, and proteomics showed the effects of maternal obesity on placental lipid transport and metabolism [86,87]. Altered lipid transport and metabolism of the obese placenta, as reflected by the changes in fatty acid transporters expression, had adverse effects on smooth placental functioning in transport and the metabolism of lipids across the feto-placental unit [88,89,90]. The obese placenta had high total lipids, triglycerides, free fatty acids, and cholesterol levels. Obese placental phenotype favours excess lipid storage with reduced lipid transport to the developing fetus, including long-chain polyunsaturated fatty acids (LCPUFAs) essentially required for fetal brain development [85]. Optimal PUFA is critical for feto-placental development, and any changes, as in obesity, can have adverse effects on fetal brain development [91,92].

The pro-inflammatory cascade is favoured by an increased ratio of pro-inflammatory M1 over anti-inflammatory M2 macrophages during maternal obesity. In addition, obesity may further enhance pathological pregnancies, such as pre-eclampsia, by reducing the uterine natural killer (uNK) cell populations [93]. Furthermore, maternal obesity was associated with epigenetic dysregulations in leptin and adiponectin secretions [94]. Thus, dysregulated endocrine controls in the term placenta deprive protective effects of these adipokines on placental development, indicating the placenta’s adaptation to a harmful maternal environment. Maternal obesity and gestational diabetes mellitus (GDM) also affect fatty acid transport across the placenta. Increased placental fatty acid binding protein 4 (FABP4) and endothelial lipase expression were observed in obese women with diabetes [95,96]. In contrast, reduced levels of FABP5 and lowered uptake of n-6 LCPUFAs were reported in obese placentas [68,87,97]. Both low and high expressions of fatty acid translocase CD36/FAT were observed in the placenta of obese women [90,97]. 

Inflammation and metabolic dysfunction, as observed in obesity, increase placental oxidative, endoplasmic reticulum stress and downstream activation of the placental unfolded protein response, all of which have been associated with pregnancy complications, such as fetal growth restriction, pre-eclampsia, and gestational diabetes. Amounts of estradiol and progesterone concentrations in plasma and placenta are reduced in obese women than in lean women [98]. The obese placenta is exposed to high insulin in early pregnancy, which produces altered steroid hormones in mitochondria and affects energy metabolism. 

Maternal lipid transport and metabolism regulate fetal adiposity via placental function. The placental transport of maternal lipids is compromised during pathological states such as IUGR and GDM. The inefficient placental LCPUFAs transfers and fat-soluble vitamins may induce metabolic dysfunction and decreased fetal growth. In GDM, the interplay of the placental ANGPTL4-lipoprotein lipase is responsible for fetal adiposity [99]. 

## 4. Maternal Lipid Transport and Metabolism and Fetal Brain Development 

The brain comprises PUFAs, especially long-chain PUFA, in which DHA and ARA constitute a major fraction of LCPUFAs. DHA levels are relatively higher in the neuronal membrane (15–50% of total fatty acids) than ARA (2–5%). Neuronal membrane fluidity facilitates neurotransmitters and synaptic membrane receptors to communicate and transmit signals among the nerve cells [100]. DHA is involved in multiple neurodevelopmental events, including neuronal differentiation [101], neuritogenesis [102], synaptogenesis [103], neurite outgrowth [104], synthesis of neuroprotective metabolites [105]. In obese pregnancy, the n-3 PUFA deficiency state may influence the endogenous synthesis of LCPUFAs from their precursors. The endogenous production of DHA and ARA is critically essential as these two fatty acids constitute major fractions of n-3 and n-6 LCPUFA for the brain and body development of the offspring. [106,107]. Placental delivery of DPA increased in mice at the expense of decreased DHA transfer during n-3 PUFA deficiency. [83]. Moreover, low n-3 PUFA-fed mice promoted FATP mRNA expression in the low placenta.

Maternal obesity is correlated with higher birth weight and is usually associated with n-3 fatty acid deficiency [108]. In utero n-3 PUFA deficiency programs the fetal brain growth and maturation and reduces neuronal and behavioural plasticity in adulthood [109]. The DHA is the brain’s predominant structural component, critically required for membrane fluidity, receptor function, and neuronal signalling [110]. As the brain depends on the DHA supply by the mother, therefore, deficiency of DHA (as happens in obesity or a high-fat diet) during brain development in the third trimester possibly affects the maturation and plasticity of the brain and its performing functions during adult life [110,111]. Arachidonic acid,20:4n-6 (ARA), quantitatively next to DHA, is required for the rapidly growing brain. Dietary n-3 fatty acid deficiency reduces DHA-regulated PLA2 and calcium-independent iPLA2 while upregulating the ARA-selective calcium-dependent cytosolic cPLA2 and secretory sPLA2 in the frontal cortex of rodents [112]. 

In addition, the deficiency of n-3 PUFA increases the expression of TNF-a and lowers glutamate receptors in the central nervous system [113]. Moreover, decreased DHA, n-6 DPA concentration and reduced telencephalon structure were observed in n-3 PUFA-depleted mice hippocampus [114]. Brain performance, such as learning, motor skills, and monoamine transmission, were affected during n-3 PUFA deficiency [115]. Maternal DHA deficiency may result in gender-specific offspring’s brain development since the efficiency of endogenous DHA conversion enzymes differs in males and females. Maternal DHA deficiency affects the offspring’s stress response, anxiety, [116] and brain reward activities [117]. The deficiency induces hypomyelination in the developing brain, predisposing the offspring to acquire anxiety-related disorders [116,117,118]. 

The free fatty acids uptake of maternal plasma is mediated by intracellular and transmembrane proteins such as FAT/CD36, FATPs, FABPpm, and FABPs in placental trophoblasts [85,119]. Dysregulated placental fatty acid transport carries increased risks of impaired neurodevelopment [120] and cardiometabolic risks [121] in the offspring. In addition, offspring born to gestational diabetic mothers reflected reduced DHA in their cord blood and showed altered rhythmic sleep maturation [122].

DHA-fed rats showed increased BDNF, GluR2, NR2B, and TrkB expression in rat brains might promote enhanced memory in rats [109,123]. Collectively, evidence suggests DHA stimulates gene expression directly or by modulating transcription factors of several membrane-associated mediators in brains that may regulate learning and memory functions. In addition, the deficiency of n-3 PUFA resulted in altered dopamine transmission in the brain, probably deranged neurogenesis [124]. However, more work is required to relate maternal obesity and omega-3 deficient conditions in pregnancy and their impact on fetal brain development. Figure 1 shows the effects of adipose lipids and metabolic hormones in pregnancies complicated by obesity on fetal brain structure and function.

## 5. Maternal Gut Microbiome and Their Impacts on Fetal Brain Development

Fetuses do not develop in a sterile microenvironment, as evidenced by a distinct bacterial signature in the placenta. The placental microbiome of obese pre-pregnant women has lesser abundance and diversity than women of normal pre-pregnancy weight. The microbial richness also reduces within the placenta from the maternal to the fetal side [125]. The 16SrRNA sequencing of gut, oral and placental microbiomes of the same pregnant women suggested that the placental microbiome has a closer similarity with the pregnant oral microbiome after cross-examining the oral non-pregnant microbiome. The study showed that the placental microbiome was enriched by several pathways, including fatty-acid metabolism [126]. The oral microflora remained stable during pregnancy but exhibited distinct composition or abundance compared to the non-pregnant state. Recent evidence suggests that oral-to-gut and gut-to-oral microbial transmission can regulate disease pathogenesis, confirming the presence of the oral-gut microbiome axis [127]. Moreover, oral microflora during pregnancy is modulated by pathological states, including GDM and PE. However, the changes in oral microbiota during pregnancy, its association with maternal health and its implications for birth outcomes are yet to be reported [128].

The microbiome is an essential functional modulator of the brain and behaviour [129]. Microbial colonization of the GI tract starts early in birth and matures towards adult composition in three years, closely parallel with brain development. Lifestyle factors like mode of delivery (vaginal or cesarean) or microbial contamination determine the basal composition of gut microbiota in the neonate. Early postnatal life is a critical determinant for continued brain development, where infants may have altered gut microbiota depending on antibiotic use, breastfeeding or formula feeding. Two major experimental approaches, antibiotic-induced dysbiosis or germ-free animals, established the gut-brain axis’s relationship. Germ-free mice studies highlighted the key role of the gut microbiota in early brain development [130,131]. The impact of maternal gut microbiota on embryonic development highlighted its role in shaping the neurometabolic system of the offspring. During pregnancy, maternal microbiota produced short-chain fatty acids (SCFAs) such as butyric acid, propionic acid, and acetic acid that determine the neuronal, intestinal and pancreatic differentiation via embryonic sensing mediated by G protein-coupled receptors, GPR43 and GPR41 [132]. A region-specific change in neurotransmitter systems was noted in the brains of germ-free mice. Germ-free mice have markedly increased 5-HT concentrations in the hippocampus [133]. The germ-free mice had upregulated genes involved with brain plasticity and metabolism, including long-term synaptic potentiation and cyclic adenosine 5-phosphate-mediated signalling system [134]. The microbiome modulates the serotonergic system in early life. A decreased hippocampal expression of the 5-HT_1A_ receptor gene in the dentate gyrus of female germ-free animals was reported [135]. Brain-derived neurotrophic factor (BDNF) is an essential plasticity-modulator protein that promotes neuronal development, growth, and survival and contributes to memory, learning, and behaviour. Expression of the BDNF gene was reduced in the cortex and amygdala in germ-free mice [134]. However, the expression of BDNF levels in the hippocampus was inconsistent in germ-free mice [133,135]. Germ-free animal data showed that the microbiome also regulates post-natal neurogenesis [136]. However, post-weaning microbial colonization of germ-free mice could not reverse the changes in adult hippocampal neurogenesis. This suggests a vital window exists in the pre-weaning period during which the microbiota regulates hippocampal neurogenesis [136]. The administration of antibiotics depleted the microbiota with decreased neurogenesis in adult animals. The antibiotic effect was reversed by physical activity or consumption of a probiotic cocktail [136]. The use of antibiotics is negatively associated with the expression of hippocampal BDNF, and the recognition memory of mice as antibiotic use decreased the gut microbial diversity and population in infants [137]. The cognitive deficit was associated with reduced bacteria-derived metabolites in the colon, altered lipid composition and changing the expression of neuronal signalling receptors such as NMDA 2B, tight junction protein, etc.

The gut microbiome does not only affect brain development per se but also alters hippocampus and amygdala function as well. The amygdala is a critical brain region, a key node for gating anxiety, fear-related response and social behaviour [138]. Germ-free (GF) mice brain shows increased amygdala volume and dendritic hypertrophy in the basolateral amygdala (BLA). The structural and functional alterations of the amygdala are associated with neuropsychiatric and developmental disorders ranging from anxiety [139] to autism spectrum disorder [140]. Moreover, germ-free mice have pyramidal BLA neurons, characterized by stubby, thin, and mushroom spines that mice with normal microbiota [139]. The differentially expressed genes, exon utilization and RNA edits were found in the amygdala of GF mice. The expression of immediate early response genes such as Fosb, Fos, and Egr2 was increased in the GF of the amygdala with concomitant increased signalling of the transcription factor CREB [141]. In germ-free mice, a significant downregulation of genes related to the immune system was reported [141]. This reflects the underdeveloped immune system and immature microglia [142]. The immune system plays a key role in mediating the microbiota’s effects on brain function.

The gut microbiome also critically regulates the pre-frontal cortical myelination [143]. GF mice had hypermyelination and upregulated expression of genes involved in myelination and myelin plasticity events in the pre-frontal cortex [143]. Administration of antibiotics during early development in rats did not affect cognitive function, immune or stress-related responses, or anxiety but observed visceral hypersensitivity in their later life [144]. The latter was associated with changes in the spinal expression of pain-associated genes [144]. The post-weaning depletion of the gut microbiota by antibiotics showed a relative change in anxiety and cognitive deficits, as observed in GF mice [145]. Moreover, the depletion reduced anxiety, induced cognitive deficits, changed tryptophan metabolic dynamics, and decreased BDNF, vasopressin, and oxytocin expression in the adult brain [145].

The gut microbiota impacts obesity as the specific microbes can induce excessive energy extraction and storage from ingested nutrients [146,147,148]. In addition, gut microbiota modulates host lipid metabolism [147], and immunity [149], and thus may promote an aberrant and chronic low-grade inflammation, as observed in obesity [148].

The gestational microbiome regulates fetal growth, which may impact brain function and adult mental cognition. Indeed, the gut microbiome is involved in mediating adverse effects of maternal challenges, such as high-fat diet exposure, obesity [150], immune activation [151], and psychosocial stress [152,153] on neurobehavioral abnormalities in mice. In addition, the latest studies indicated that the maternal microbiome modulates host responses to acute insults in the brain, but it is not known whether it impacts offspring brain development.

The maternal gut microbiome promotes fetal thalamocortical axonogenesis by signalling microbe-modulating metabolites to develop neurons in the brain [20]. “Dysbiosis” of the maternal gut microbiome, in response to a high-fat diet [150], stress [153], and infection [151] during pregnancy, is associated with abnormal brain function and behaviour in the offspring [129]. Manipulating the maternal microbiome and its metabolites during pregnancy produce offspring with altered tactile sensitivity in two aversive somatosensory behavioural tasks, with no overt differences in many other sensorimotor behaviours. The gut microbiota modulates numerous bioactive compounds in the intestine, blood and various other organs [154]. The maternal gut microbiota regulates metabolites in the maternal and fetal tissues, including the fetal brain. The maternal gut microbiota-regulated fetal brain metabolites can modulate axon outgrowth in thalamic explants of mice and promote fetal thalamocortical axonogenesis in offspring. Microbiome metabolites trimethylamine-N-oxide, imidazole propionate, N, N, N-trimethyl-5-aminovalerate, 3-indoxyl sulfate and hippurate are involved with neurological status and neurite outgrowth [20], although the molecular mechanisms of actions of these microbial metabolites are still not known. A poorly developed maternal microbiome was associated with decreased brain white matter in the adult offspring [20,155,156,157]. Inflammation-induced changes in the maternal gut microbiome disrupted somatosensory cortical architecture in adult mouse offspring [158].

The microbiomes in malnourished children showed dysregulated expression of axonogenesis proteins, which were alleviated by treatment with microbiota-enriched diets [159]. Epidemiological data also associated maternal infection and antibiotic use with a greater risk for neurodevelopmental complications in the offspring [19,160]. Whether alterations may mediate these effects in the maternal microbiome of human remain unclear. The interactions between the gut microbiome and fetal nervous system begin prenatally through influences of the maternal gut microbiota on fetal brain metabolomic profiles and gene expression. However, new studies are required to identify whether early to mid-gestation is a critical period during which the maternal microbiome promotes fetal neurodevelopment to support developmental processes. Figure 2 describes the maternal microbiota in obesity exerts effects on the neonatal brain and development.

## 6. Conclusions

The entero-mammary axis enables mothers to transfer microbes from the gut to the mammary gland. While breastmilk is reported to influence gut microbiota, gut mucosal immunity and adipose development [161], no data is available on fetal brain development yet. Obesity in pregnancy causes metabolic syndrome with altered inflammation status, endocrine factors, placental dysfunction, and maternal gut dysbiosis. Consequently, the placental transport of essential LCPUFAs for fetal brain development during the last trimester is disturbed. All these factors contribute to neurodevelopmental alterations in offspring via different mechanisms. The maternal gut microbiota changes fetal brain metabolites and thus affects the brain development and function of the offspring. The important role of the maternal microbiome in offspring neurodevelopment is now increasingly appreciated. Altogether, obesity and maternal microbiome modulate fetal neurodevelopment processes that may influence later-life behaviours.

## Figures and Tables

**Figure 1 nutrients-14-04515-f001:**
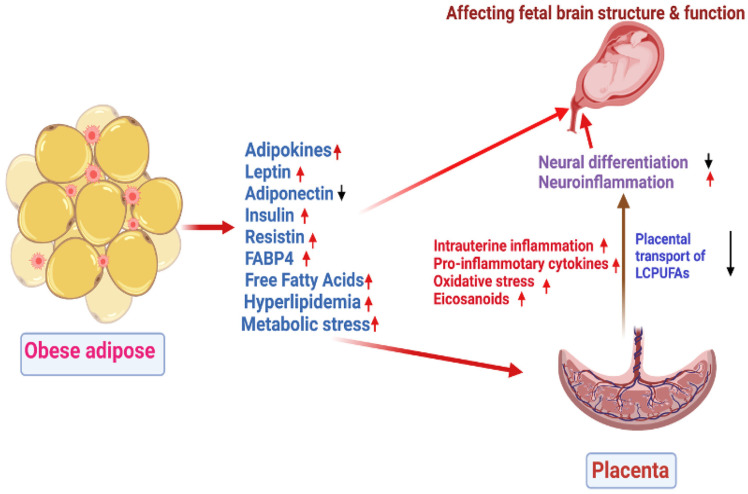
Diverse adipokines, cytokines, FABP4 and metabolic stress in obese women may affect fetal brain development. In pregnancies complicated by obesity, these factors mediate crosstalk among different cell populations within adipose and travel to remote organs to regulate their function. The long-term neurodevelopmental programming is linked to inflammation and placental serotonin production and LCPUFAs supply. Red arrows indicate “upregulation” while black arrows indicate “downregulation.

**Figure 2 nutrients-14-04515-f002:**
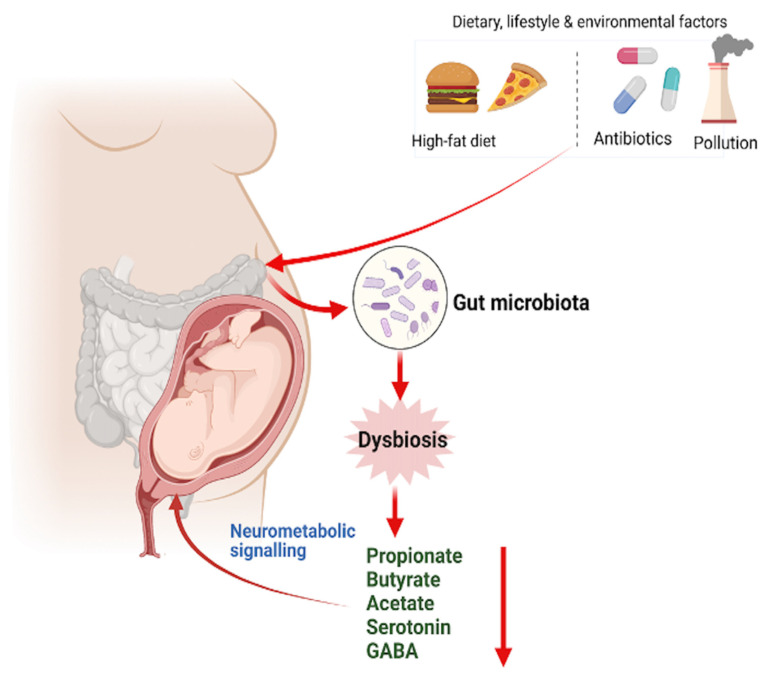
Maternal gut microbiota is modulated by obesity, antibiotics and a high-fat diet. Gut dysbiosis can affect fetal brain development via different mechanisms. Since, at birth, the newborn is colonized by maternal and environmental microbiota, therefore maternal gut microbe also affects the neonatal gut microbe. The gut microbiota signal to the adult brain via the vagus nerve, bacterial metabolites, gut hormones, and immune signalling. More data has now emerged that these pathways are functional in the fetus and newborn. Red arrows indicate “upregulation” while black arrows indicate “downregulation.

**Table 1 nutrients-14-04515-t001:** Epidemiological studies on maternal obesity and neurodevelopment of the offspring.

Observations	References
Increased odds of developing autism spectrum disorders in offspring of obese.	[52,53,54,55]
Increases odds of cognitive deficits in children of obese women observed in cohorts.	[55,56,57,58,59,60]
Obese mothers have twice as likely to have a child with mental developmental delay.	[58,61]
An increase in autism in children was reported in prospective pregnancy cohorts	[62]
The maternal obesity was associated with schizophrenia in adult offspring in a large retrospective cohort study but other studies could not confirm this association.	[63,64]
A dose-dependent increase in relative risk of cerebral palsy as maternal BMI was observed.	[13,65,66,67]

## Data Availability

Not Applicable.

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
