# Peer review of "Maternal Obesity and Gut Microbiota Are Associated with Fetal Brain Development"

_nutrients, 2022, doi:10.3390/nu14214515_

Round 1

Reviewer 1 Report

Maternal obesity and gut microbiota are associated with fetal brain development 

Main observations:

The manuscript topic is consistent with the journal content. However, the main conclusions about a high-fat diet in the pregnant woman (in Abstract, Summary) are inconsistent with the content presented in the manuscript and references to the subject literature. The Western diet - with a poor-balanced proportion of particular types of unsaturated fatty acids (of cis or trans configuration) and saturated fatty acids is the main problem in fetal brain development, not a high-fat diet.

Need precision in nomenclature, e.g., should be used "VEGF" instead of "VEGFA" and should be used "LCPUFAs" instead of "LCPUAs."

LACK of LIMITATION of study (at the end of the Discussion section), for example , that most of the study is conducted with an animal model.

Literature is relatively out of date - more than 22% are articles more than ten years old - more everyday items should be used.

I believe this study would be a candidate for publication in your journal as a review article with significant revisions.

Abbreviations used in the manuscript should be clarified the first time they are used. 

For example, instead of:

Obesity and a high-fat diet in pregnancy can induce placental and intrauterine inflammation and thus influence the neurodevelopmental outcomes of the offspring.

Should be:

Obesity and a diet with acid fat o configuration trans in pregnancy can induce placental and intrauterine inflammation and thus influence the neurodevelopmental outcomes of the offspring.

Instead of:

Chronic activation of glucocorticoid receptors alters the levels of glucocorticoid in hippocampal and hypothalamic regions, thus modifying feedback regulation of the HPA axis [47].

Should be:

Chronic activation of glucocorticoid receptors alters the levels of glucocorticoid in hippocampal and hypothalamic regions, thus modifying feedback regulation of the hypothalamic-pituitary-adrenal (HPA) axis [47].

Instead of:

The n-3 PUFA deficiency resembles high-fat diet-induced impaired placental phenotypes. 

Should be:

The n-3 polyunsaturated fatty acids (n-3 PUFA) deficiency resembles high-fat diet-induced impaired placental phenotypes. 

Instead of:

Optimal placentation is facilitated by several angiogenic factors such as vascular endothelial growth factor A (VEGFA), ANGPTL4, FGF, and PlGF, as well as docosahexaenoic acid, 22:6n-3 (DHA) [62,63].

Should be:

Optimal placentation is facilitated by several angiogenic factors such as vascular endothelial growth factor (VEGF), angiopoietin-like 4 (ANGPTL4), fibroblast growth factors (FGF), and placental growth factor (PIGF), as well as docosahexaenoic acid, 22:6n-3 (DHA) [62,63].

Instead of:

Obese placental phenotype favours excess lipid storage with reduced lipid transport to the developing fetus, including LCPUFAs essentially required for fetal brain development [69]. 

Should be:

Obese placental phenotype favours excess lipid storage with reduced lipid transport to the developing fetus, including long-chain polyunsaturated fatty acids (LCPUFAs) essentially required for fetal brain development [69]. 

Instead of:

The long-term neurodevelopmental programming is linked to inflammation and placental serotonin production and LCPUAs supply.

Should be:

The long-term neurodevelopmental programming is linked to inflammation and placental serotonin production and LCPUFAs supply.

Author Response

Open Review

(x) I would not like to sign my review report  

 ( ) I would like to sign my review report  

English language and style

( ) Extensive editing of English language and style required  

 ( ) Moderate English changes required  

 (x) English language and style are fine/minor spell check required  

 ( ) I don't feel qualified to judge about the English language and style  

  Is the work a significant contribution to the field?

  Is the work well organized and comprehensively described?

  Is the work scientifically sound and not misleading?

  Are there appropriate and adequate references to related and previous work?

  Is the English used correct and readable?

Comments and Suggestions for Authors

Maternal obesity and gut microbiota are associated with fetal brain development 

Main observations:

  1. The manuscript topic is consistent with the journal content. However, the main conclusions about a high-fat diet in the pregnant woman (in Abstract, Summary) are inconsistent with the content presented in the manuscript and references to the subject literature. The Western diet - with a poor-balanced proportion of particular types of unsaturated fatty acids (of cis or trans configuration) and saturated fatty acids is the main problem in fetal brain development, not a high-fat diet.

Response: Thank you for pointing out this issue. The work primarily characterized the impact of obesity during pregnancy and its effects on brain development. The physiological state of pregnancy was the major impetus of the manuscript. Therefore, the “obesity” state is highlighted in both the abstract and Summary. In addition, the diverse dietary lipids were discussed because of their contribution to obesity.

  1. Need precision in nomenclature, e.g., should be used "VEGF" instead of "VEGFA" and should be used "LCPUFAs" instead of "LCPUAs."

Response: The cited references (62 and 63) reported VEGFA subtypes. LCPUAs are changed to LCPUFAs.

  1. LACK of LIMITATION of study (at the end of the Discussion section), for example, that most of the study is conducted with an animal model.

Response: The review article collected evidence from animal and clinical studies. “No data is available on fetal brain development” is mentioned in the discussion.

  1. Literature is relatively out of date - more than 22% are articles more than ten years old - more everyday items should be used.

I believe this study would be a candidate for publication in your journal as a review article with significant revisions.

Response: Original and accurate citations were prioritized in this manuscript, including a fine blend of old and new references.

  1. Abbreviations used in the manuscript should be clarified the first time they are used. 

For example, instead of:

Obesity and a high-fat diet in pregnancy can induce placental and intrauterine inflammation and thus influence the neurodevelopmental outcomes of the offspring.

Should be:

Obesity and a diet with acid fat o configuration trans in pregnancy can induce placental and intrauterine inflammation and thus influence the neurodevelopmental outcomes of the offspring.

Response: Thank you. We have revised it now.

  1. Instead of:

Chronic activation of glucocorticoid receptors alters the levels of glucocorticoid in hippocampal and hypothalamic regions, thus modifying feedback regulation of the HPA axis [47].

Should be:

Chronic activation of glucocorticoid receptors alters the levels of glucocorticoid in hippocampal and hypothalamic regions, thus modifying feedback regulation of the hypothalamic-pituitary-adrenal (HPA) axis [47].

Response: Thank you. We have revised it now.

  1. Instead of:

The n-3 PUFA deficiency resembles high-fat diet-induced impaired placental phenotypes. 

Should be:

The n-3 polyunsaturated fatty acids (n-3 PUFA) deficiency resembles high-fat diet-induced impaired placental phenotypes. 

Response: Thank you. We have revised it now.

  1. Instead of:

Optimal placentation is facilitated by several angiogenic factors such as vascular endothelial growth factor A (VEGFA), ANGPTL4, FGF, and PlGF, as well as docosahexaenoic acid, 22:6n-3 (DHA) [62,63].

Should be:

Optimal placentation is facilitated by several angiogenic factors such as vascular endothelial growth factor (VEGF), angiopoietin-like 4 (ANGPTL4), fibroblast growth factors (FGF), and placental growth factor (PIGF), as well as docosahexaenoic acid, 22:6n-3 (DHA) [62,63].

Response: Thank you. We have revised it now.

  1. Instead of:

Obese placental phenotype favours excess lipid storage with reduced lipid transport to the developing fetus, including LCPUFAs essentially required for fetal brain development [69]. 

Should be:

Obese placental phenotype favours excess lipid storage with reduced lipid transport to the developing fetus, including long-chain polyunsaturated fatty acids (LCPUFAs) essentially required for fetal brain development [69]. 

Response: Thank you. We have revised it now. 

  1. Instead of:

The long-term neurodevelopmental programming is linked to inflammation and placental serotonin production and LCPUAs supply.

Should be:

The long-term neurodevelopmental programming is linked to inflammation and placental serotonin production and LCPUFAs supply. 

Response: Thank you. We have revised it now. 

Open Review

(x) I would not like to sign my review report  

 ( ) I would like to sign my review report  

English language and style

( ) Extensive editing of English language and style required  

 ( ) Moderate English changes required  

 ( ) English language and style are fine/minor spell check required  

 (x) I don't feel qualified to judge about the English language and style  

  Is the work a significant contribution to the field?

  Is the work well organized and comprehensively described?

  Is the work scientifically sound and not misleading?

  Are there appropriate and adequate references to related and previous work?

  Is the English used correct and readable?

Comments and Suggestions for Authors

The authors reviewed the effects of maternal obesity and gut microbiota on fetal neurodevelopment outcomes in four different parts. 

  1. Impacts of obesity on maternal endocrine factors and fetal brain development
  2. The placenta of an obese mother and its impact on fetal brain development
  3. Maternal lipid transport and metabolism and fetal brain development
  4. Maternal gut microbiome and their impacts on fetal brain development

The Figure 1 and Figure 2 provided good Summary for part 3 and 4 and gave the readers a good overview. However, this review is largely descriptive. Suggest authors to provide similar summary Figures for part 1 and 2 as well to help readers to catch the main points.

  1. Response: The overview and the summary figure are designed in the Abstract figure (please refer to)
  2. The references in Table 1 are not within the order of the text next to it. Please check. 

Response: Table 1 provided information in the context that associates maternal obesity and neurodevelopment in the offspring. Citation ordering by Endnote software needs to be reviewed.

Reviewer 2 Report

The authors reviewed the effects of maternal obesity and gut microbiota on fetal neurodevelopment outcomes in four different parts. 

1. Impacts of obesity on maternal endocrine factors and fetal brain development

2. The placenta of an obese mother and its impact on fetal brain development

3. Maternal lipid transport and metabolism and fetal brain development

4. Maternal gut microbiome and their impacts on fetal brain development

The Figure 1 and Figure 2 provided good summary for part 3 and 4 and gave the readers a good overview. However, this review is largely descriptive. Suggest authors to provide similar summary Figures for part 1 and 2 as well to help readers to catch the main points.

The references in Table 1 are not within the order of the text next to it. Please check. 

Author Response

(The authors gave the same response as above.)

Round 2

Reviewer 1 Report

Overall Recommendation

  -  Accept in present form

Reviewer 2 Report

The authors reviewed the effects of maternal obesity and gut microbiota on fetal neurodevelopment outcomes in four different parts. 

  1. Impacts of obesity on maternal endocrine factors and fetal brain development
  2. The placenta of an obese mother and its impact on fetal brain development
  3. Maternal lipid transport and metabolism and fetal brain development
  4. Maternal gut microbiome and their impacts on fetal brain development

The Figure 1 and Figure 2 provided good Summary for part 3 and 4 and gave the readers a good overview. However, this review is largely descriptive. Still think it is necessary for  authors to provide similar summary Figures for part 1 ad 2 as well to help readers.

Suggest authors to provide a list of abbreviations at the beginning or the end of the manuscript as well.